# The Contribution of Lipidomics in Ovarian Cancer Management: A Systematic Review

**DOI:** 10.3390/ijms241813961

**Published:** 2023-09-11

**Authors:** Vasiliki Tzelepi, Helen Gika, Olga Begou, Eleni Timotheadou

**Affiliations:** 1Department of Oncology, “Papageorgiou” General Hospital, 56429 Thessaloniki, Greece; timotheadou@auth.gr; 2Biomic_Auth, Bioanalysis and Omics Lab, Centre for Interdisciplinary Research of Aristotle University of Thessaloniki, Innovation Area of Thessaloniki, 57001 Thermi, Greece; gkikae@auth.gr (H.G.); olina_18@hotmail.com (O.B.); 3Laboratory of Forensic Medicine and Toxicology, School of Medicine, Aristotle University of Thessaloniki, 54124 Thessaloniki, Greece; 4Department of Chemistry, Aristotle University of Thessaloniki, 54124 Thessaloniki, Greece; 5Department of Medical Oncology, Aristotle University of Thessaloniki School of Medicine, 54124 Thessaloniki, Greece

**Keywords:** ovarian cancer, diagnostic, prognostic, predictive biomarker, lipidomics

## Abstract

Lipidomics is a comprehensive study of all lipid components in living cells, serum, plasma, or tissues, with the aim of discovering diagnostic, prognostic, and predictive biomarkers for diseases such as malignant tumors. This systematic review evaluates studies, applying lipidomics to the diagnosis, prognosis, prediction, and differentiation of malignant and benign ovarian tumors. A literature search was performed in PubMed, Science Direct, and SciFinder. Only publications written in English after 2012 were included. Relevant citations were identified from the reference lists of primary included studies and were also included in our list. All studies included referred to the application of lipidomics in serum/plasma samples from human cases of OC, some of which also included tumor tissue samples. In some of the included studies, metabolome analysis was also performed, in which other metabolites were identified in addition to lipids. Qualitative data were assessed, and the risk of bias was determined using the ROBINS-I tool. A total of twenty-nine studies were included, fifteen of which applied non-targeted lipidomics, seven applied targeted lipidomics, and seven were reviews relevant to our objectives. Most studies focused on the potential application of lipidomics in the diagnosis of OC and showed that phospholipids and sphingolipids change most significantly during disease development. In conclusion, this systematic review highlights the potential contribution of lipids as biomarkers in OC management.

## 1. Introduction

OC is one of the most common gynecologic cancers, ranking eighth among women worldwide, with most cases having a poor prognosis due to late diagnosis. When diagnosed early (stage I/II), the 5-year overall survival rate is 92%, compared with only about 29% when diagnosed late (stage III/IV) [1]. Early detection is difficult due to its non-specific symptoms: early satiety, bloating, changes in bowel habits, weight loss, fatigue, and frequent urination [2].

Even though serum Cancer Antigen 125 (CA125) is considered the “gold standard” biomarker for OC, it has low sensitivity in the early stages (I–II), as it is only elevated in 26–50% of stage I diseases and has low specificity (sometimes leading to false positives) during menstruation and conditions such as fibroids and endometriosis. Another serum biomarker, HE4 (human epididymis protein 4), was found to be over-expressed in 93% of serous and 100% of endometrioid OC cases. However, even if its specificity is higher when compared to CA125, several factors such as age, smoking, and use of hormonal contraceptives cause great variations in its levels [3].

To date, it is well known that lipids play a critical role in proliferation, survival, death, and interactions between cells due to their involvement in cell membranes, cellular signaling, cellular interactions, and energy storage. These processes are also related to the transformation, progression, and metastasis of cancer; thus, several lipids may be mediators in oncogenic processes and could be potential cancer biomarkers [4].

Lipidomics, first employed in 2003 [5], is a subdiscipline of metabolomics, where the investigation of the quantitative and qualitative profiles of lipid components in biofluids, tissues, cells, and organisms takes place [6]. It is a rapidly evolving scientific field due to technological advances and it seems that, along with other fields, it could contribute to the characterization, detection, and classification of cancerous cells and tissues, as well as to the differentiation between normal and neoplastic environments. Thus, the integration of lipidomics into cancer research could be applied to the discovery of new biomarkers, contributing to the diagnosis, prognosis, and prediction of several malignancies [4].

In the last decade, studies regarding the application of lipidomics in OC management have been emerging. To date, no systematic review has been published exploring the contribution of lipidomics in OC. However, Salim et al. (2020) conducted a systematic review that evaluated the contribution of metabolomics in OC, where some lipids were also identified alongside other small molecules [7]. Therefore, the aim of this systematic review is to evaluate the literature regarding the potential application of lipidomics in the diagnosis, prognosis, and prediction of OC, and to discriminate between malignant and benign ovarian tumors.

## 2. Methods

### 2.1. Eligibility Criteria

In order to focus on the applicability of lipidomics in OC management, we have included only human studies in our systematic review. In particular, studies that were deemed eligible for the review were focused on lipidomic analyses of blood, in particular serum or plasma from pre-surgery cases, while those involving only tissues or cell line samples were excluded. The main criterion of our systematic search was to find studies regarding lipid alterations during the development and progression of all subtypes of EOC. Thus, all studies regarding lipidomic analysis of high-grade serous (the most common subtype of EOC), low-grade serous, mucinous, endometrioid, and clear cell carcinoma subtypes were included. In order to investigate the most recent scientific achievements on this topic, studies carried out in the last decade were included. Publications not written in English were deemed ineligible. Our review focused on non-targeted lipidomics. A supplementary analysis, consisting of seven targeted lipidomics studies in OC, was included to confirm the consistency of untargeted lipidomics.

### 2.2. Search Strategies

The systematic search followed PRISMA guidelines (Preferred Reporting Items for Systematic reviews and Meta-Analyses) [8]. A bibliographic search was conducted on PubMed, Science Direct, and SciFinder between January 2021 and 2023. The augmentation of our search consisted of references cited in primary sources (Figure 1). Keywords used in our research were “ovarian cancer”, “diagnostic/prognostic/predictive biomarker”, and “lipidomics”. Two authors worked independently on the bibliographic search. After the search by the first author (VT), based on inclusion and exclusion criteria, a total of 32 studies were collected and screened. The whole process was repeated independently by a second author (OB). Communication for obtaining or confirmation of data from study authors was not conducted. The screening process is summarized in the flow diagram (Figure 1). Collection of relevant data from primary reports was conducted by the first reviewer (VT), cross-checked by the second (OB), and the whole screening process was cross-checked by two senior authors (ET, EG). Information and data collected included the year of publication, author details, country and aim of the study, the kind and size of the sample, analytical methodology and technique used, findings, and conclusions. The main characteristics of the included non-targeted and targeted studies are summarized in Table 1 and Table 2, respectively.

### 2.3. Study Risk of Bias Assessment

The ROBINS-I tool (Risk Of Bias In Non-randomized Studies of Interventions) [9] was used for the assessment of the methodological quality of all studies, except relevant reviews. The two co-authors worked independently during this process. Each study was assigned as “low”, “moderate”, “serious”, or “critical” risk of bias or “no information”. Results from the methodological assessment are summarized in Table 3 and Table 4 for untargeted and targeted lipidomics, respectively. In order to assess the consistency of results from untargeted lipidomic studies, we conducted a supplementary analysis of seven targeted lipidomic studies using the ROBINS-I tool, the results of which are summarized in Table 4.

### 2.4. Synthesis of the Results

Table 1 and Table 2 display all of the pooled data on the included studies in our systematic review, consisting of 15 non-targeted and 7 targeted lipidomic studies.

**Table 1 ijms-24-13961-t001:** Pooled data from non-targeted lipidomic studies.

Author, Date, Country	Sample & Method	Aim of Study	Study Findings	Conclusions
Iurova et al. (2022) Russia [10]	CC Total (*n* = 41) HGSC (*n* = 28): Stage I-II (*n* = 5) Stage III-IV (*n* = 23) HC (*n* = 13) Pre-operative plasma samples HPLC-QTOF-MS	D	Decreased plasma concentration of Plasmanyl-LPC(O-16:0), Plasmenyl-PE(P-18:0/18:2,18:0/20:3,18:0/20:4, 18:1/22:6), Plasmenyl-PC(P-16:1/18:0), LPC(14:0,17:0,18:2), PS(37:5), SM(d20:0/18:4) and CerNS(d18:1/24:0) in patients with early OC.	Lipid profiling by HPLC-MS can improve identification of early-stage OC and thus increase the efficiency of treatment
Salminen et al. (2021) Germany, Finland [11]	CC Total (*n* = 711) Turku (Finland): Malignant (*n* = 197) Benign (*n* = 114) Charite 1 (Germany): Malignant (*n* = 51) Charite 2 (Germany): Malignant (*n* = 104) Charite 3 (Germany): Malignant (*n* = 147) Benign (*n* = 98) Pre-treatment serum samples LC-QTRAP	P	A two-lipid signature, based on the ratio of the ceramide Cer(d18:1/18:0) and the phosphatidylcholine PC-O(38:4) identified especially poor-outcome patients at the time of diagnosis, before any oncological treatment	The two-lipid signature was able to identify EOC patients with an especially poor prognosis at the time of diagnosis, showing promise for the detection of disease relapse.
Wang et al. (2021) China [12]	CC Total (*n* = 340) Discovery set (*n* = 153) EOC (*n* = 62): Early (*n* = 25) Advanced (*n* = 37) BOT (*n* = 41) HC (*n* = 50) Validation set (*n* = 187): EOC (*n* = 47) BOT (*n* = 29) HC (*n* = 39) Else (*n* = 72) Serum samples UPLC-QTOF-MS	D	Compared to HC, levels of FAs, LPCs and LPEs were significantly increased, while PCs, PC-Os, PE-Os, SMs and PIs were decreased in BOT and EOC. All of SFAs, MUFAs and PUFAs in BOT and EOC presented a considerable increase simultaneously. PC-O 36:2 (FA18:2), PC-O 38:3 (FA20:3), PC-O 38:4 (FA20:4), PE (16:0p 18:1) and PE (18:0p 22:5) presented lower levels in EEOC vs. AEOC. Levels of TGs were remarkably decreased in BOT, compared with HC. Levels of Cers and FAs increased, and levels of PCs, PC-Os and PE-Os decreased in EOC, compared with HC. Cers, DGs, PCs, PEs, and TGs presented high level in EOC, while PC-Os and PE-Os presented low level in EOC, compared to BOT. Cer(d18:1/16:0), PC-O(36:2), PE(16:0p_18:1), OAHFA(18:2_24:6) were selected as the combinational diagnostic marker.	This study provided evidence for the mechanistic understanding of OC at the level of lipid metabolism. The defined potential combinational marker would be helpful for aiding EOC diagnosis, especially for early stage EOC.
Buas et al. (2021) USA [13]	CC Total (*n* = 218) 1st cohort (FH) (*n* = 100) OC (*n* = 50) BOT (*n* = 50) 2nd cohort (RP) (*n* = 118) OC (*n* = 60) BOT (*n* = 58) Pre-operative Plasma samples Ascites samples (*n* = 15) Direct infusion MS	D	In both cohorts, reductions in TAG, PC, CE species, DAG, PE, LPC, LPE and SM species were observed in patients vs. controls, while CER(18:0) was elevated. Differentially abundant lipid species in Early stage (I/II): TAG, DAG, PC, PE Late stage (III/IV): TAG, CE, PC, PE, LPC, LPE, SM, CER. PE, LPC, LPE exhibited significant reductions in cases vs. controls. In stage-stratified analyses, certain significant class-level differences were detected only in late-stage (LPC, LPE, SM), only in early-stage (DAG, TAG) or in both subgroups (PC, PE). Results suggest that combining CA125 with specific individual lipid metabolites, such as DAG(16:1/18:1), may provide a substantial boost to specificity at 90% sensitivity, relative to CA125 alone, in separating early-stage ovarian malignancies from benign adnexal masses. Certain metabolites may exhibit changes in circulation years before an OC diagnosis: PCs (34:2, 38:3), PEs (36:3, 38:5), TAGs (46:0, 52:4/5, 54:4/5/6/7, 56:4/5/7/8), and SM 22:1	Potential translational utility of specific circulating lipid metabolites to aid in the clinical diagnosis and triage of women with adnexal mass.
Niemi et al. (2018) Germany, Finland [14]	CC Total (*n* = 604) Malignant (*n* = 290): Charite (*n* = 62) Finland (*n* = 76) Charite discovery (*n* = 152) Borderline (*n* = 25): Charite (*n* = 18) Finland (*n* = 7) Benign (*n* = 289): Charite (*n* = 109) Finland (*n* = 82) Charite discovery (*n* = 98) Pre-operative serum/plasma samples Non-targeted & Targeted lipidomics UPLC-QTRAP	D	Patients showed a consistent decrease in the concentration of most of phospholipids (PCs, LPCs, PIs), cholesteryl esters, glucosyl/galactosyl Cers, and sphingomyelins. Cers with 18:0, 20:0 and 24:1 FAs were increased, while 24:0 FA-containing Cers were decreased. TAGs with shorter FA side chains were decreased, whereas those with longer FA side chains were increased. Twenty-one of 23 lipids analyzed were decreased in all histological subtypes, and only Cer(d18:1/18:0) and TAG (18:1/18:1/20:4) were increased. It appears that borderline tumors do not cause as much of a change to the lipidome as malignant tumors. Lipids improved diagnostic value of CA125 for the detection of stage I/II	Changes in lipid metabolism due to OC occur in early-stage disease but intensify with increasing stage. Understanding lipid metabolism in OC may lead to new therapeutic and diagnostic alternatives
Braicu et al. (2017) Germany [15]	CC Total (*n* = 245) Pre-operative serum samples: HGSC (*n* = 147) Controls (*n* = 98) Pre-operative tissue samples: HGSC (*n* = 140) LC-QTOF-MS	D, P	OC patients exhibit decreased serum levels of PCs, PEs, PIs, CEs, DAGs, SMs, cerebrosides (Glc/GalCers), LacCers, Gb3s, S1Ps. Cers, with 16:0, 18:0, 20:0 and 24:1 FAs were increased, while those containing 23:0 and 24:0 FAs were decreased. TAGs with short FA side chain were decreased, while long chain TAGs were the same or increased compared to controls. The predictive value of diagnosis was improved by the combination of CA125 with PEO-36:1. Lipids belonging to the CE, SM, LPC, PC, PC O and PE O lipid classes were decreased in all OC patients and progressed to lower levels especially in patients where the whole macroscopic tumor could not be removed during the surgery. Ceramides elevated in cancer patients continued to increase during disease progression. Cer(d18:1/16:0) showed significant hazard ratio both in overall and progression-free survival analyses.	Alterations in lipid metabolism in OC could contribute to diagnosis and prognosis of the disease
Xie et al. (2017) China [16]	Prospective study Median follow-up (37.5 months) Total (*n* = 98) Dead in 3 years (*n* = 46) Survived after 3 years (*n* = 52) Pretreatment plasma samples UPLC-QTOF-MS	Pr	Poor survival with the increase of Kynurenine, Acetylcarnitine and PC(42:11) and with the decrease of LPE(22:0/0:0). The 4 potential predictive biomarkers were significantly altered in short-term mortality compared to long-term survival patients (*p* < 0.05). PC (42:11) and LPE(22:0/0:0) were significantly altered in short-term mortality and medium survival. Patients with long-term survival showed increased plasma relative intensity of LPE (20:0/0:0) and decreased relative intensity of PC(42:11)	Plasma metabolites could be utilized to predict the overall survival and discriminate the short-term mortality and long-term survival for EOC patients
Li et al. (2017) China [17]	Prospective study Total (*n* = 70) EOC recurrent (*n* = 39): ER (*n* = 12) LR (*n* = 27) Non recurrent (*n* = 31): Pre-operative plasma samples UPLC-QTOF-MS	Pr	Most of the identified lipids in EOC recurrent patients were decreased compared with the non-recurrent ones, except upregulated PC(31:2) and PE-P(42:4). LysoPG(20:5), as a potential biomarker, could provide an AUC value of 0.736, significantly increasing the predictive power of clinical characteristics from AUC value 0.739 to 0.875. Decreased LysoPG(20:5) level was identified as the most important prognostic feature. LysoPCs were downregulated in recurrent EOC patients compared with the non-recurrent patients. A series of PCs were downregulated in EOC recurrent patients. Cer(d18:1/23:0), SM(d18:1/14:0), SM(d18:2/14:0) were decreased in EOC recurrent patients. PIs levels were lower in patients with recurrent EOC than in those without recurrent EOC. Decreased levels of TGs: a specific metabolic feature for early relapse.	Plasma lipidomics study could be used for predicting EOC recurrences, as well as early and late recurrent cases. The lipid biomarker research improves the predictive power of clinical predictors, and the identified biomarkers are of great prognostic and therapeutic potential
Buas et al. (2016) USA [18]	CC Total (*n* = 100) Serous OC (*n* = 50) Serous BOT (*n* = 50) Plasma samples at the time of surgery Non-targeted & Targeted lipidomics LC-QTOF-MS	Discr	Glycerolipids and glycerophospholipids were found to be decreased in abundance in cases relative to controls	Alterations in circulating plasma lipid metabolites are associated with the presence of malignant ovarian carcinoma versus benign ovarian tumor.
Ke et al. (2016) China [19]	CC Total (*n* = 105) Primary EOC (*n* = 35) The same Post-operative EOC (*n* = 35) Relapsed EOC (*n* = 35) Controls (*n* = 35) Plasma samples UPLC-QTOF-MS	P, Pr	Compared with controls, significantly lower concentrations of tetracosahexaenoic acid, 2-octenoic acid, 12,13-DiHODE and 19,20-DiHDPA were observed in primary EOC. Post-operative EOC patients had increased fatty acids and decreased LPCs. Significantly increased levels of LPCs, LPEs and fatty acids were seen in EOC recurrent patients	There are delineated metabolic changes in response to advanced EOC, surgery and recurrence, and identified biomarkers that could facilitate both understanding and monitoring of EOC development and progression
Y. Zhang et al. (2016) China [20]	CC Total (*n* = 65) OC (*n* = 27) BOT (*n* = 27) HC (*n* = 11) Plasma samples UPLC-QTOF-MS	D	LPCs were upregulated and PCs and TGs were downregulated in OC patients compared to Benign and Healthy controls. (Potential biomarkers: 16:0 LPC, 18:1 LPC, 20:3 LPC, 20:4 LPC, 22:6 LPC, 16:0/18:1 PC, 16:0/18:2 PC, 18:0/18:2 PC, 18:0/20:5 PC, 18:2/18:2 PC, 18:2/18:2/16:0 TG)	MS-based lipidomics is a powerful method in discovering new potential clinical biomarkers for diseases.
Hou et al. (2016) China [21]	CC Total (*n* = 215) EOC (*n* = 139) BOT (*n* = 38) UF (*n* = 38) Pre-operative plasma samples UPLC-QTOF-MS	D, Pr	All the GPs were decreased in EOC patients vs. controls, except PC(33:5) and PC(34:3). SPs were remarkably increased in EOC patients, except SM(d18:2/14:0). Two types of glycerolipids showed the opposite trend in EOC patients: MG were significantly increased, whereas DG were significantly decreased in EOC patients vs. BOT/UF. All the PCs and pPEs were negatively associated with pathological stage, except PC(33:5), and SMs and Cers were positively associated with pathological stages, except SM(d18:2/14:0). MG was positively associated, whereas DG was negatively associated with pathological staging. PC(P-38:4), PC(35:5), PC(34:3), SM(d18:1/17:0) and SM(d18:0/16:1) together with CA125 improved the diagnostic and predictive accuracy of CA125.	Plasma lipid profiles analyzed by UPLC- QTOF/MS could be used to discriminate EOC from controls. Promising lipid metabolites together with CA125 improved the diagnostic and predictive performance and accuracy of EOC.
Gaul et al. (2015) USA [22]	CC Total (*n* = 95) EOC (I/II) (*n* = 46) HC (*n* = 49) Serum samples Non-targeted & targeted lipidomics UPLC-HRMS	D	16 metabolites were found to have optimal accuracy in distinguishing between early-stage EOC and controls when used in a linear support vector machine model, most of which were lipids and fatty acids, including lysophospholipids: LPE and LPI	The results provide the foundation of clinically significant diagnostic tests and evidence for the importance of alterations in lipid and fatty acid metabolism in the onset and progression of the disease.
Zhang et al. (2015) China [23]	Prospective study Total (*n* = 38) EOC (III/IV) (*n* = 38): With recurrence (*n* = 26) Without recurrence (*n* = 12) Pre-treatment plasma samples UPLC-QTOF-MS	P/Pr	Metabolites identified as potential metabolic biomarkers of EOC recurrence: L-tryptophan (AUC = 0.80), LysoPC(14:0) (AUC = 0.77) and LysoPE(18:2) (AUC = 0.82) decreased in EOC patients with recurrence, whereas kynurenine (AUC = 0.79) and bilirubin (AUC = 0.76) increased. Patients with and without recurrent EOC could be distinguished using this panel of metabolites (AUC = 0.91).	Remarkably, combining of these five biomarkers provided an AUC value of 0.91, which suggests strong potential for predicting EOC recurrence.
T. Zhang et al. (2012) China [24]	CC Total (*n* = 170) Training samples: EOC (*n* = 50): BOT (*n* = 50) External validation samples: EOC (*n* = 30) BOT (*n* = 40) Pre-operative plasma samples UPLC-QTOF-MS	Discr	The plasma L-Tryptophan, LysoPC(18:3), LysoPC(14:0), and 2-Piperidinone concentrations were lower among EOC patients than those among BOT patients, either in the training set or in the external validation set	UPLC-QTOF/MS based metabolomic platform possessed a favorable value in discriminating malignant from benign ovarian tumors.

CC = Case-Control, HGSC = High Grade Serous Carcinoma, HC = Healthy Controls, HPLC-MS = High Performance Liquid Chromatography Mass Spectrometry, D = Diagnosis, LPC = Lyso-phosphatidyl-choline, PE = Phosphatidyl-ethanolamine, PC = Phosphatidyl-choline, SM = Sphingomyelin, CER = Ceramide, P = Prognosis, EOC = Epithelial Ovarian Cancer, BOT = Benign Ovarian Tumour, FAs = Fatty Acids, PI = Phosphatidyl-inositol, SFAs = Saturated Fatty Acids, MUFAs = Mono-unsaturated Fatty Acids, PUFAs = Poly-unsaturated Fatty Acids, EEOC = Early EOC, AEOC = Advanced EOC, TG = Triacyl-glycerol, DG = Diacyl-glycerol, OAHFA = (O-acyl)-ω-hydroxy fatty acid, LPE = Lyso-phosphatidyl-ethanolamine, ER = Early Recurrent, LR = Late Recurrent, Discr = Discrimination, Pr = Prediction, UF = Uterine Fibroid.

**Table 2 ijms-24-13961-t002:** Pooled data from targeted lipidomic studies.

Author, Date, Country	Sample & Method	Aim of Study	Study Findings	Conclusions
Hishinuma et al. (2021) Japan [25]	CC Total (*n* = 160) EOC (*n* = 80) HC (*n* = 80) Plasma samples UHPLC-MS/MS	D, Pr	Decreased concentrations of LPCs and PCs and increased concentrations of TGs were observed in EOC patients compared to HCs	Plasma metabolome analysis is useful not only for the diagnosis of EOC, but also for predicting prognosis.
Yagi et al. (2020) USA [26]	CC Total (*n* = 62) OC (*n* = 20) BOT (*n* = 20) HC (*n* = 22) Plasma samples HPLC-MS	D	LPE (22:6)/LPE (o-16:0) has the best sensitivity in distinguishing between control and benign, SM (d18:1/24:1)/SM (d18:1/22:0) has the best sensitivity between control and cancer, and PE (16:0/18:1)/PE (o-18:0/18:2) has the best specificity in distinguishing between benign and cancer.	Potential of plasma phospholipids as a novel marker of OC with great sensitivity and specificity by utilizing the unique characteristics of phospholipids to further enhance the diagnostic power.
Zeleznik et al. (2020) USA [27]	Nested CC Total (*n* = 504) Mean follow-up (*n* = 12.3 years) Cases (*n* = 252) Serous/PD tumors (*n* = 176) Endometrioid/CC (*n* = 34) Rapidly fatal tumors (*n* = 86) Controls (*n* = 252) Pre-diagnosis plasma samples LC-MS/MS	A	The top three metabolites associated with risk were pseudouridine, C18:0 sphingomyelin (SM) and 4-acetamidobutanoate. Differential association by acyl carbon number and double bond content of TAGs with risk of OC overall was observed. Specifically, TAGs with higher number of acyl carbon atoms and double bonds were associated with increased risk, while TAGs with lower number of acyl carbon atoms and double bonds were associated with decreased risk. Fifty-three lipid-related metabolites (26TAGs, 7PCs, 6LPEs, 3PEs, 3LPC, 4DAGs, 2LPSs, and 2PSs) showed differences by tumor aggressiveness.	This study suggests that TAGs may be important as a novel risk biomarker for OC, particularly for rapidly fatal tumors, with associations differing by structural features.
Plewa et al. (2019) Poland [28]	CC Total (*n* = 76) OC (*n* = 26) BOT (*n* = 25) HC (*n* = 25) Serum samples HPLC-TQ/MS	D	Decreased serum levels of LPC a C16:1, PC aa C32:2, PC aa C34:4 and PC aa C 36:6 in OC patients compared to BOT and HCs	There is dominant role of lipid alterations in OC.
Kozar et al. (2018) Slovenia [29]	CC Total (*n* = 57) EOC (*n* = 15) BOT (*n* = 21) HC (*n* = 21) Pre-treatment serum samples HPLC-TQ/MS	D	Five most significant markers were Cer 34:1;2 (C16), Cer 40:1;2 (C22), Cer 42:1;2 (C24), SM 36:0;2 and SM 36:1;2 (C18 and C18:1). Important increase in levels of C16-Ceramide, long chain Cers C22/C24 and in C18 and C18:1 Sphingomyelin levels in EOC vs. Controls were observed	Long chain ceramides and sphingomyelins may serve as a novel biomarker for EOC detection and may also offer insight into the role of sphingolipid metabolism in cell proliferation.
Knapp et al. (2017) Poland [30]	CC Total (*n* = 155) AOC (*n* = 74) HC (*n* = 81) Pre-operative Plasma samples LC-MS/MS Post-surgery Tissue samples UHPLC/MS/MS	Pr	Significant increase (higher risk of OC) in C16-Cer (>311.88 ng/100 μL), C18:1-Cer (>4.75 ng/100 μL) and C18-Cer (>100.76 ng/100 μL) was noticed in plasma of AOC patients vs. controls. Increase in C16-Cer, C18:1-Cer, C18-Cer, C24:1-Cer, C24-Cer and S1P was noticed in ovarian tissue of AOC women compared to controls	Some sphingolipids can be used as potential biomarkers of advanced ovarian cancer and they can play an important role in the pathogenesis of this disease
Shan et al. (2012) USA [31]	CC Total (*n* = 423) EOC (*n* = 211): Stage I/II (*n* = 78) Stage III/IV (*n* = 133) BOT (*n* = 212) Pre-operative serum samples LC-ESI-MS/MS	D	The additional measurement of LPA, PPE, LPC (14:0, 12:0) supplements results of CA125 measurement and improves diagnostic accuracy. Measurement of phospholipids improved the identification of early-stage cases from 65% (based on CA125) to 82%, and for mucinous cases from 44% to 88%	Measurement of specific biologically active phospholipids improves diagnostic sensitivity and accuracy among women with suspected ovarian cancer

PD tumors = Poorly Differentiated tumors, A = Agnostic, AOC = Advanced Ovarian Cancer.

**Table 3 ijms-24-13961-t003:** Risk of bias assessment in non-targeted lipidomic studies.

Study ID	Study Type	Pre-Intervention		At-Intervention	Post-Intervention				Total Score
		Confounding Bias	Selection Bias	Classification Bias	Deviation Bias	Missing Data Bias	Measurement of Outcome Bias	Selective Reporting Bias	Overall Risk of Bias Judgement
Iurova et al. (2022) [10]	CC	S	M	L	L	L	L	L	S
Salminen et al. (2021) [11]	CC	M	M	L	L	L	M	L	M
Wang et al. (2021) [12]	CC	S	L	L	L	L	L	L	S
Buas et al. (2021) [13]	CC	M	L	L	L	L	L	L	M
Niemi et al. (2018) [14]	CC	M	M	L	L	L	L	L	M
Braicu et al. (2017) [15]	CC	M	L	L	L	L	L	L	M
Xie et al. (2017) [16]	PS	M	L	L	L	L	L	L	M
Li et al. (2017) [17]	CC	M	L	L	L	L	L	L	M
Buas et al. (2016) [18]	CC	M	M	L	L	L	L	L	M
Ke et al. (2016) [19]	CC	M	M	L	L	L	L	L	M
Y. Zhang et al. (2015) [20]	CC	S	M	L	L	L	M	L	S
Hou et al. (2015) [21]	CC	M	L	L	L	L	M	L	M
Gaul et al. (2015) [22]	CC	S	M	L	L	L	M	L	S
H.Zhang et al. (2014) [23]	PS	M	L	L	L	L	L	L	M

CC = Case-Control, PS = Prospective Study, M = Moderate, L = Low, S = Serious.

**Table 4 ijms-24-13961-t004:** Risk of bias assessment in targeted lipidomic studies.

Study ID	Study Type	Pre-Intervention		At-Intervention	Post-Intervention				Total Score
		Confounding Bias	Selection Bias	Classification Bias	Deviation Bias	Missing Data Bias	Measurement of Outcome Bias	Selective Reporting Bias	Overall Risk of Bias Judgement
Hishinuma et al. (2021) [25]	CC	M	L	L	L	L	L	L	M
Yagi et al. (2020) [26]	CC	S	L	L	L	L	S	L	S
Zeleznik et al. (2020) [27]	CC	M	M	M	L	L	L	L	M
Plewa et al. (2019) [28]	CC	M	L	L	L	M	L	L	M
Kozar et al. (2018) [29]	CC	M	L	L	L	L	M	M	M
Knapp et al. (2018) [30]	CC	M	L	L	L	L	L	L	M
Shan et al. (2012) [31]	CC	M	L	L	L	L	M	L	M

## 3. Results

### 3.1. Study Selection and Characteristics

Our initial search consisted of 32 papers regarding untargeted lipidomics in OC, of which 24 were excluded due to being in Russian (*n* = 1), being published before 2012 (*n* = 10), and including cell lines/tissue samples (*n* = 13). The included studies were all relevant to the lipidomic analysis of serum/plasma. Additionally, seven abstracts were included for evaluation after a careful search of primary source citations. A supplementary analysis, consisting of seven targeted lipidomic analyses in OC, was included in our review to enhance the validity of untargeted lipidomic analyses. Useful information from seven relevant reviews were included for confirmation of our results. The whole screening process can be found in Figure 1.

In total, 22 studies of untargeted lipidomics (*n* = 15) and targeted lipidomics (*n* = 7) were included in our review, consisting of 4552 participants, 2504 of whom were diagnosed with OC and 2048 who served as controls (benign ovarian tumor/healthy controls). Ranging from 38 to 711, the median sample size was 206. Most studies were conducted in China (*n* = 8), followed by the USA (*n* = 6). The other studies were conducted in Germany (*n* = 3), Finland (*n* = 2), Poland (*n* = 2), Slovenia (*n* = 1), Russia (*n* = 1) and Japan (*n* = 1). The reviews included in our study were conducted in the USA (*n* = 3), Australia (*n* = 2), and Italy (*n* = 1), and one was a co-operation of the following countries: China, USA, Austria, Poland, Finland, UK, France, Canada, Germany, Slovenia, and Korea.

### 3.2. Risk of Bias in Studies

As reflected by outcomes of the ROBINS I tool assessment, the majority of included retrospective studies were open to several sources of bias since 4 out of 15 untargeted (Table 3) and 1 out of 7 targeted lipidomic studies (Table 4) were classified as being at serious risk of bias [10,12,20,22,26].

Studies by Iurova et al. [10], Niemi et al. [14], Hou et al. [21], Zhang H. et al. [23], Zhang T. et al. [24], and Kozar et al. [29] excluded comorbidities from their participants, such as liver, kidney, and metabolic diseases, as well as medication, attempting to reduce selection bias, while in the other studies, comorbidities were not excluded or were not mentioned at all.

Any significant difference in the level of lipids between patients with OC and controls should be carefully evaluated as it is almost impossible for a non-randomized observational study to eliminate all types of bias.

It should be noted that the small sample size in some studies conducted in Russia (Iurova et al. [10]), Slovenia (Kozar et al. [29]), and Poland (Plewa et al. [28]) can be explained by the low prevalence of ovarian cancer in these populations. However, the smaller the sample size of a study, the lower its power, and the smaller the size of the effect evaluated, such as differences in the level of several lipids in serum/plasma between OC patients and controls, the larger the sample size required. Out of the 22 studies included in our review, nine had a sample size smaller than 100, but even in those with more than 100 participants, it is possible that they were not adequately powered due to a lack of pilot data that could guide the right choice of sample size. The study of Salminen et al. [11] had the largest sample size, where a total of 711 participants (*n* = 499 OC patients and *n* = 212 BOT) from four cohorts took part in the experimental measurements.

In “omics” evaluations, such as metabolomics and lipidomics, it is common to observe the introduction of false discoveries due to multiple scenarios being tested and the comparison of several lipids in two or more groups (OC patients and BOT and healthy controls in our review). Therefore, it is important that studies included the false discovery rate (FDR) [32]. In our review, studies that used the FDR were those of Wang et al. [12], Buas et al. [13,18], Zeleznik et al. [27], Ke et al. [19], Hou et al. [21], and Plewa et al. [28].

Studies by Wang et al. [12], Buas et al. [13,18], Niemi et al. [14], and Zhang T. et al. [24] included a discovery cohort in order to evaluate lipid differences and predictive models, decreasing selection bias. All other included studies cross-validated predictive models from the same cohort.

The basic target of studies included in our review was significant differences in lipidomic profile between OC patients and controls (BOT and/or healthy controls). Most studies used unsupervised principal component analysis (PCA), which is capable of revealing the separation of data between groups. Through this approach, potential bias can be revealed [33].

Most of our studies used the ultra-performance liquid chromatography-mass spectrometry (UPLC-MS) technique either for acquiring a comprehensive lipidomic profile of patients’ and controls’ serum/plasma or for targeting specific predefined lipids. Results could be affected by various parameters, such as type of specimen (serum or plasma), analytical protocol applied and instrumentation used for analysis, processing software and spectral libraries used for identification, and statistical analysis. However, this limitation cannot be avoided in “omics” studies as harmonization in workflows is not yet put in place.

### 3.3. Results of Syntheses

#### 3.3.1. Diagnosis

In total, 13 out of 22 studies included in this review demonstrated the application of lipidomics in the diagnosis of OC. Eight of them concerned non-targeted lipidomics [10,12,13,14,15,20,21,22] and the other five were targeted studies [25,26,28,29,31]. Data, findings, and conclusions of these studies can be found in Table 1 and Table 2 for untargeted and targeted lipidomics, respectively. Lipids, identified to have significantly different levels in serum/plasma between OC patients and BOT or healthy controls, are summarized in Table 5 and Table 6 for untargeted and targeted lipidomic analyses. In particular, in non-targeted lipidomic studies, upregulation of ceramides (Cers) with 18:0, 20:0, and 24:1 fatty acyls (FAs) and triacylglycerols (TAGs) with longer FA side chains in OC patients was reported [12,13,14,15], which was also confirmed by targeted analyses [29] while downregulation of phosphatidylcholine (PC) [12,13,15,20], sphingomyelin (SM) [10,12,13,14,15], and diacylglycerols (DGs) [13,15,21] was noted. Cers, containing 23:0 and 24:0 FAs and TAGs with short FA side chains, was reported as downregulated in OC patients [14,15]. In targeted lipidomic studies, downregulation of lysophosphatidylcholine (LPC) was reported in EOC patients versus controls [25,28,31].

**Table 5 ijms-24-13961-t005:** Upregulated/downregulated lipids in non-targeted lipidomic analyses.

Non-Targeted Study	Up-Regulated Lipids	Down-Regulated Lipids
Iurova et al. (2022) [10]		Plasmanyl-LPC(O-16:0), Plasmenyl-PE (P18:0/18:2, 18:0/20:3, 18:0/20:4, 18:1/22:6), Plasmenyl-PC(P-16:1/18:0), LPC(14:0,17:0,18:2), PS(37:5), SM(d20:0/18:4) and CerNS(d18:1/24:0) in patients vs. controls
Salminen et al. (2021) [11]	Cer(d18:1/18:0)	PC-O(38:4)
Wang et al. (2021) [12]	FFAs, LPCs and LPEs in EOC/BOT vs. controls. Cers in EOC vs. controls	PCs, PC-Os, PE-Os, SMs and PIs in EOC/BOT vs. controls
Buas et al. (2021) [13]	CER(18:0) in patients vs. controls	TAG, PC, CE species, DAG, PE, LPC, LPE and SM species in patients vs. controls
Niemi et al. (2018) [14]	Cers with 18:0, 20:0 and 24:1 FAs, TAGs with longer FA side chains	Phospholipids (PCs, LPCs, PIs), cholesteryl esters, glucosyl/galactosyl Cers, and sphingomyelins, 24:0 FA-containing Cers, TAGs with shorter FA side chains
Braicu et al. (2017) [15]	Cers, with 16:0, 18:0, 20:0 and 24:1 FAs. Long chain TAGs.	PCs, PEs, PIs, CEs, DAGs, SMs, cerebrosides(Glc/GalCers), LacCers, Gb3s, S1Ps. CERs containing 23:0 and 24:0 FAs. TAGs with short FA side chain.
Li et al. (2017) [17]	PC(31:2) and PE-P(42:4) in EOC recurrent patients	LysoPCs, PCs, PIs in recurrent EOC patients compared with the non-recurrent patients. LysoPG(20:5), Cer(d18:1/23:0), SM(d18:1/14:0), SM(d18:2/14:0), TGs in EOC recurrent
Xie et al. (2017) [16]	Plasma relative intensity of LPE (20:0/0:0) in patients with long-term survival	Relative intensity of PC(42:11) in patients with long-term survival
Buas et al. (2016) [18]		Glycerolipids and glycerophospholipids in cases versus controls
Ke et al. (2016) [19]	Fatty acids in Post-operative EOC patients. LPCs, LPEs and fatty acids in EOC recurrent patients.	LPCs, Tetracosahexaenoic acid, 2-octenoic acid, 12,13-DiHODE and 19,20-DiHDPA of primary EOC patients compared to controls.
Y.Zhang et al. (2016) [20]	LPCs in patients compared to BOT and controls	PCs and TGs in patients compared to BOT and controls
Hou et al. (2016) [21]	All the GPs in EOC patients vs. controls, except PC(33:5) and PC(34:3). MG and SPs in EOC patients, except SM(d18:2/14:0)	DG in EOC patients vs. BOT/UF
Gaul et al. (2015) [22]	LPE and LPI in OC patients vs. controls	
H. Zhang et al. (2015) [23]		LysoPC(14:0) and LysoPE(18:2) decreased in EOC patients with recurrence compared to non-recurrent
T. Zhang et al. (2012) [24]		LysoPC(18:3), LysoPC(14:0) levels were lower in EOC patients compared to BOT patients

**Table 6 ijms-24-13961-t006:** Up/Down-regulated lipids in targeted lipidomic analyses.

Non-Targeted Study	Up-Regulated Lipids	Down-Regulated Lipids
Hishinuma et al. (2021) [25]	TGs in EOC vs. HCs	LPCs and PCs in EOC vs. HCs
Yagi et al. (2020) [26]		
Zeleznik et al. (2020) [27]		
Plewa et al. (2019) [28]		LysoPC a C16:1, PC aa C32:2, PC aa C34:4 and PC aa C 36:6
Kozar et al. (2018) [29]	C16-Ceramide, long chain Ceramides C22/C24 and C18 and C18:1 Sphingomyelin levels in EOC vs. Controls	
Knapp et al. (2017) [30]	C16-Cer, C18:1-Cer and C18-Cer in AOC patients compared to controls	
Shan et al. (2012) [31]		PPE, LPC (14:0, 12:0) in OC patients compared to controls

#### 3.3.2. Prognosis

Four untargeted lipidomic studies in our review studied the prognosis of OC [11,15,19,23]. Data, findings, and conclusions of these studies can be found in Table 1, and the lipids that were found to be altered in patients compared to controls in Table 5. Analysis by Zhang H et al. [23] demonstrated that LysoPC(14:0) and LysoPE(18:2) were decreased in EOC patients with recurrence compared to non-recurrent, while Salminen et al. [11] and Braicu et al. [15] agree that upregulation of Cers could be useful in predicting OC prognosis. Moreover, Ke et al. [19] reported that LPCs and LPEs were upregulated in EOC-recurrent patients.

#### 3.3.3. Prediction

From a total of 22 studies, included in this review, five untargeted and two targeted lipidomic/metabolomic analyses investigated the prediction of OC [16,17,19,21,23,25,30]. Data, findings, and conclusions of these studies can be found in Table 1 and Table 2 for non-targeted and targeted studies, respectively, and lipids identified to be altered in patients versus controls can be found in Table 5 and Table 6. In untargeted analyses, Li et al. [17] and H. Zhang et al. [23] reported that specific LysoPC and LysoPE species decreased in EOC patients with recurrence compared to non-recurrent, while Ke et al. [19] reported that some others were elevated. Furthermore, Xie et al. [16] demonstrated that plasma-relative intensity of LPE (20:0/0:0) was upregulated in patients with long-term survival, while the relative intensity of PC (42:11) in patients with long-term survival was downregulated. Hou et al. [21] and the targeted analysis of Hishinuma et al. [25] reported upregulation of glycerophospholipids as a potential predictive biomarker for OC, whereas Knapp et al. [30] suggested upregulation of several Cers for further investigation as biomarkers for the prediction of the disease.

#### 3.3.4. Discrimination between OC and BOT

Two untargeted lipidomic studies referred to the contribution of “omics” to discrimination between malignant and benign ovarian tumors [18,24]. Data, findings, and conclusions of these studies can be found in Table 1, and lipids altered in OC versus BOT in Table 5. Buas et al. [18] recommended that glycerolipids and glycerophospholipids found to be downregulated in cases versus controls should be further investigated for their possible application in discrimination between OC and BOT, while T. Zhang et al. [24] refer to decreased LysoPC(18:3) and LysoPC(14:0) levels in EOC patients compared to BOT patients. Finally, Zeleznik et al. [27] conducted a prospective untargeted/targeted analysis of circulating plasma metabolites (lipids included) in order to find a potential correlation with OC risk. Specifically, TAGs with a higher number of acyl carbon atoms and double bonds were associated with increased risk, while TAGs with a lower number of acyl carbon atoms and double bonds were associated with decreased risk. Fifty-three lipid-related metabolites (26TAGs, 7PCs, 6LPEs, 3PEs, 3LPC, 4DAGs, 2LPSs, and 2PSs) showed differences in tumor aggressiveness. Data regarding this study can be found in Table 2.

## 4. Discussion

Ovarian cancer, known as the “silent killer”, is a heterogeneous disease. Its origin can be epithelial (90% of cases) or non-epithelial (5% germ cell and 5% sex cord stromal tumors). There are several differences in their epidemiology, etiology, and therapeutic choices. The deadliest type of OC is epithelial ovarian cancer (EOC), for which there is no reliable, effective screening test. As a result, these patients are usually diagnosed at advanced stages, with an estimated 5-year overall survival of 20–40%. Histologically, EOC can be divided into five main subtypes: high- and low-grade serous (75–80%), endometrioid (10%), clear cell carcinoma (10%), and mucinous (3%). EOC patients usually respond well to cytoreductive surgery with preoperative or adjuvant platinum-based chemotherapy. However, the estimated progression-free survival (PFS) is approximately 18 months [1,2]. Therefore, it is of great importance to discover and develop diagnostic, prognostic, and predictive biomarkers in order to diagnose the disease in the early stages and detect which patients have better prognoses and respond to novel therapies such as bevacizumab and Poly-ADP-Ribose (PARP) inhibitors [2].

One of the scientific fields that could contribute to this direction is the investigation of lipid metabolism and their alterations during carcinogenesis and progression of OC by lipidomic analyses since it is already known that several lipids a play critical role during normal and cancerous cell life [4]. To date, seven reviews regarding the involvement of small molecules, lipids included, in the development of OC have been published [4,7,34,35,36,37,38]. The first one was that of Pyragius et al. [37], where it was reported that OC cells depend on lipids for their increased energy requirements during tumor growth, with phospholipids and sphingolipids being the main lipid classes implicated in carcinogenesis. In accordance with this review, reviews of Kreitzburg et al. in 2018 [35] and Pitman et al. in 2021 [34] agreed and enhanced the idea that the S1P pathway (Sphingosine-1-Phosphate), which involves ceramides, sphingomyelins, and spingosine-1-phosphate, is implicated in oncogenic processes such as proliferation, migration, neo-vascularization, and metastasis. Combining these conclusions with the results of our systematic literature search, it is obvious that glycerophospholipid metabolism could be considered a key dysregulated metabolic pathway in OC.

Our systematic review is the third one to investigate the implications of lipid metabolites in OC, following Turkoglu et al. in 2016 [36] and Salim et al. in 2020 [7], which also explored other metabolites, apart from lipids, through metabolomic analyses in serum, plasma, tissue, or cell lines of OC cells. Many of the studies included in our review were mentioned in Turkoglu’s and Salim’s papers. The conclusion introduced by their reviews, as well as ours, is that there is strong evidence that lipid aberrations contribute to OC development and offer new insights into the pathogenesis of the disease with the potential to integrate novel diagnostic, prognostic, and predictive biomarkers to the “gold standard” biomarker CA125. Indeed, Niemi et al. [14] reported that the combination of CA125 measurements with those of significantly altered lipids, such as LPCs and PCs, improved the diagnostic capability of CA125 at early stages (I/II) of the disease. More precisely, it was noted that in early stages (I/II) of all histologic subtypes of EOC, there was a consistent decrease in the concentration of PCs, LPCs, PIs, cholesteryl esters, glucosyl/galactosyl Cers, and sphingomyelins. Moreover, Cers with 18:0, 20:0, and 24:1 FAs were increased, while those with 24:0FAs were decreased. Finally, TAGs with longer FA side chains were increased, while those with shorter side chains were decreased. Moreover, Buas et al. [13] referred to the increase in accuracy of a multivariate model, consisting of four lipid biomarkers belonging to the group of DAGs, TAGs, and CERs, to discriminate between early-stage malignant and benign ovarian masses after the integration of CA125 levels. For example, one of their models demonstrated that the combination of CA125 levels with levels of DAG (16:1/18:1) could substantially boost the specificity at 90% sensitivity compared to CA125 alone in distinguishing early-stage ovarian malignancies from benign adnexal masses.

It should be noted that Perrotti et al. [4] in their review in 2016 mentioned lysophosphatidic acid (LPA) to be significantly increased in OC patients compared to healthy controls, based on the study of Meleh et al. [39] in 2007. As knowledge and techniques have evolved, Yagi et al. [40] in 2019 analyzed potential inconsistencies in the use of LPA as a biomarker in OC due to the possible effect of storage time of samples (artificial LPA generation during incubation at room temperature) and concentration of ethylene-diamine-tetra-acetic acid (EDTA) during blood drawing. Thus, we should consider with skepticism the role of LPA as a biomarker in OC, since false positive results could lead to misdiagnosis with devastating consequences for patients and the health care system.

The possible role of upregulated long-chain ceramides and sphingomyelins as diagnostic biomarkers in OC should be more thoroughly evaluated in future studies as recorded by Salminen et al. [11], Niemi et al. [14], Kozar et al. [29], Braicu et al. [15], and Knapp et al. [30]. However, apart from congruent results in our review, there are also results that contradict each other. Indeed, most of our included studies demonstrated decreased levels of LPCs in OC patients versus BOT and healthy controls, while Zhang Y. et al. [20] displayed upregulation of LPCs. This could be attributed to the small sample size (only *n* = 65 participants) or to the population included. Furthermore, Wang et al. [12] agreed with other studies that PCs are downregulated in patients versus controls and Cers are upregulated, but they demonstrated an increase in LPCs in patients compared to controls, which is contradictory to other studies.

It should be noted that most targeted and untargeted studies included all histologic subtypes of EOC in their samples concomitantly. Iurova et al. [10] (*n* = 28 serum samples) and Braicu et al. [15] (*n* = 147 serum samples) were the only untargeted studies focused on lipid alterations exclusive to high-grade serous ovarian cancer, which is the most common subtype of EOC cases. Both studies showed a consistent decrease in serum samples of phosphatidylcholines (PCs), phosphatidylethanolamines (PEs), phosphatidylinositols (PIs), cholesteryl esters (CEs), diacylglycerols (DAGs), sphingomyelins (SMs), cerebrosides (glucosyl/galactosylceramides (Glc/GalCers), lactosylceramides (LacCers)), globotriasoylceramides (Gb3s), and sphingosine-1-phosphates (S1Ps) in OC patients compared to controls. In addition, they both agreed that ceramide alterations were dependent on the FA side chain composition. In particular, CERs with 16:0, 18:0, 20:0, and 24:1 FAs were increased in patients compared to controls, while those containing 23:0 and 24:0 FAs were decreased and TAGs with short FA chains were decreased and those with long FA chains were at the same level or increased in patients compared to controls. All other studies included samples of all histologic subtypes with most of them belonging to HGSOC (probably due to its high prevalence) and a small number of other subtypes. This could be considered a potential source of bias and it is recommended by all studies that future lipidomic analyses should include a higher number of patients. Moreover, we suggest that future studies should focus separately on each subtype of EOC due to the great heterogeneity between them.

In addition, according to the NCCN guidelines (2022) [2] for OC, bevacizumab, an anti-vascular endothelial growth factor, has been recommended as an option for first-line platinum-based chemotherapy after cytoreductive surgery, based on the results of two randomized phase III trials, GOG-0218 (2011) and ICON7 (2015). Furthermore, since 2014, olaparib, a Poly-ADP-Ribose-Polymerase inhibitor, has been integrated into the therapeutic management of ovarian cancer after FDA approval, followed by rucaparib and niraparib. None of the studies included in our review mentioned the inclusion of these new agents in patients’ therapy. Therefore, it would be of great interest to explore if lipid alterations during therapy with these new agents could become a biomarker for identifying patients who are most likely to benefit.

The most important strength of this review is that it included all studies regarding lipidomic analyses related to the contribution of lipidomic serum/plasma profiles of patients with malignant or benign ovarian tumors and healthy controls as potential biomarkers in this deadly disease. Our supplementary analysis, consisting of seven targeted studies, and the inclusion of seven former reviews, enhanced the validity and accuracy of our review. Excluding studies published before 2012 could be considered a source of bias. Including both serum and plasma samples could also affect the results. Therefore, we recommend skepticism about lipid changes that take place during OC development. Out of the 22 included studies, eight were conducted in China, while the rest were in North America and Europe. This could be considered a confounding factor that should be more thoroughly evaluated in future studies. Studies identified as being at serious risk of bias (mostly due to several confounding factors) should be considered with great skepticism.

## 5. Conclusions

Several lipid classes, such as LPCs, PCs, Cers, and TGs, seem to play a significant role in oncogenic processes contributing to the development of OC. Many studies have aimed to elucidate and evaluate their involvement in this highly heterogeneous disease. Most studies have focused on the serous subtype, which is the most common type of OC. Through literature searches, it is obvious that a great effort has been made in the last decade to identify potential lipid biomarkers for the diagnosis, prognosis, and prediction of this fatal disease. Due to several confounding factors in many of the included studies, we recommend that future studies focus on more appropriate matching between patients and controls, such as the number of patients and controls, age, menopausal status, comorbidities, medications, and levels of CA125.

We demonstrated that the incorporation of lipid biomarkers into the existing biomarkers CA125 and HE4 increased their sensitivity and specificity in the diagnosis of the disease.

Overall, our review highlighted the importance of lipid aberrations and lipid pathway alterations during the development of OC. Future untargeted and targeted lipidomic analyses, adequately designed, could elucidate oncogenic pathways involved in the pathogenesis of this malignancy of poor prognosis and improve its screening, detection, and recurrence.

## Figures and Tables

**Figure 1 ijms-24-13961-f001:**
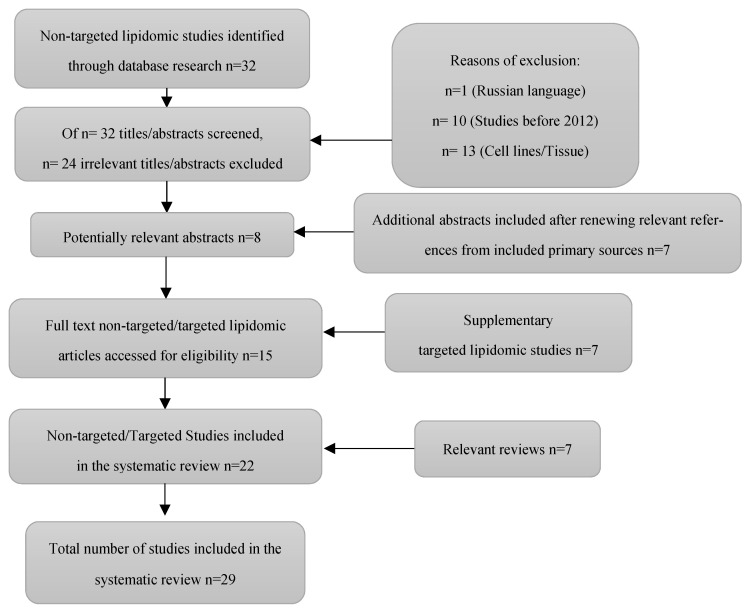
Flow diagram of the search strategy and study selection process.

## Data Availability

Not applicable.

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
