# Peer review of "The Contribution of Lipidomics in Ovarian Cancer Management: A Systematic Review"

_ijms, 2023, doi:10.3390/ijms241813961_

Round 1

Reviewer 1 Report

Discussion of the manuscript has no strength in the present form. This should be rewritten. 

Author Response

Response to Reviewer 1 Comments

Point 1: Discussion of the manuscript has no strength in the present form. This should be rewritten.

Response 1: The writing of the discussion was a very painful procedure due to the extensive data, which I tried to evaluate thoroughly and demonstrate them as simple as possible. I emphasized at the greatest lipid alterations shown within all studies, targeted and untargeted. Thus, I did not rewrite the discussion, but I added a few paragraphs and phrases, according to yours and the others reviewers’ suggestions, in order to enhance its strength. In red you can see phrases I deleted, paragraphs I added, and English language editing.

Thank you in advance.

Reviewer 2 Report

This document is not in the review format proposed by the journal (ijms-template). You can review the guidelines at:

“Submission Checklist”

Use the Microsoft Word template or LaTeX template to prepare your manuscript; (https://www.mdpi.com/journal/ijms/instructions).

I tried to adjust it to the guidelines, but the format does not allow it. I recommend authors restructure the document and resubmit it.

Greetings.

Author Response

Point 1: This document is not in the review format proposed by the journal (ijms-template). You can review the guidelines at:

“Submission Checklist”

Use the Microsoft Word template or LaTeX template to prepare your manuscript; (https://www.mdpi.com/journal/ijms/instructions).

I tried to adjust it to the guidelines, but the format does not allow it. I recommend authors restructure the document and resubmit it.

Response 1: I used the Microsoft Word template for the preparation of the manuscript as you suggested, and I resubmitted it.

Thank you in advance.

Reviewer 3 Report

I read with great interest the Manuscript titled " The Contribution Of Lipidomics In Ovarian Cancer Management: A Systematic Review”, topic interesting enough to attract readers' attention.

Although the manuscript can be considered already of good quality, I would suggest following recommendations: 

-       I suggest a round of language revision, in order to correct few typos and improve readability.

-       .It would be interesting to discuss topic and results of this study could contribute to determine more accurately thechoose of tailored therapy in ovarian cancer patients, considering recent evidence in literature. I would be glad if the authors discuss this important point, referring to PMID: 37314974 and 27794568. 

Because of these reasons, the article should be revised and completed. Considering all these points, I think it could be of interest to the readers and, in my opinion, it deserves the priority to be published after minor revisions.

  I suggest a round of language revision, in order to correct few typos and improve readability.

Author Response

Point 1:   I suggest a round of language revision, in order to correct few typos and improve readability.

Response 1: I checked the grammar and possible mistakes of the manuscript as you suggested. In red you can see all corrections.

Point 2: It would be interesting to discuss topic and results of this study could contribute to determine more accurately the choose of tailored therapy in ovarian cancer patients, considering recent evidence in literature. I would be glad if the authors discuss this important point, referring to PMID: 37314974 and 27794568.

Response 2: Even if I did not use these PMID you suggested, I added in the discussion a small paragraph (written in red) where I mention that none of our included studies had samples from OC patients under therapy with these new agents (bevacizumab or PARP inhibitors). I based my paragraph to NCCN guidelines version 2022, and I am suggesting future studies to explore the potential role of lipids as biomarkers for patients receiving the highest benefit from these novel therapies.

Thank you in advance.

Reviewer 4 Report

The review by Tzelepi and co-authors is based on the current scientific literature reporting the contribution of lipidomics in progression and management of ovarian cancer. The review is timely, as it highlights a lesser studied/reviewed topic in the field, which could be of importance for successful management of this deadly disease.

The review is well-written, and the scientific literature is comprehensively analyzed and presented.

Critique point. Although the authors briefly mention that most studies included the high grade serous histotype as the primary subject of the study, it would be helpful to indicate for each study the histotypes included and whether any differences between histotypes were found.

Author Response

Point 1: Although the authors briefly mention that most studies included the high grade serous histotype as the primary subject of the study, it would be helpful to indicate for each study the histotypes included and whether any differences between histotypes were found.

Response 1: I included a paragraph in the discussion (written in red) where I mention that only two studies focused exclusively to HGSC (the most common subtype of EOC). I considered confusing for the readers to include in my tables the number of every subtype, so I wrote the number of OC cases and controls. I did not separate the subtypes because results did not focus on each subtype separately. Thus, I recommend in the discussion that future studies should focus on each subtype separately since there is great heterogeneity between them.

Thank you in advance.

Reviewer 5 Report

The authors focus on the lipid components between malignant and benign ovarian masses via literature review. 

Major issues:

1. In the introduction part, you should include previous research about how lipid components influence the carcinogenesis of ovarian cancer.

2. You should detail the criteria for selection of papers in the methodology part.

3. You have included a lot of tables in your manuscript, and it will be better if you can replace some with figures which are more visually straightforward.

Author Response

Response to Reviewer 5 Comments

Point 1: In the introduction part, you should include previous research about how lipid components influence the carcinogenesis of ovarian cancer.

Response 1: In my introduction, in the 3d paragraph, I mention that it is well known the critical role of lipids in carcinogenic processes. There is no study regarding exclusively lipids’ role in ovarian cancer carcinogenesis.

Point 2: You should detail the criteria for selection of papers in the methodology part.

Response 2: In the paragraph entitled “Eligibility criteria” I added a phrase (written in red) where I explain the aim of our systematic search and the kind of studies included.

Point 3: You have included a lot of tables in your manuscript, and it will be better if you can replace some with figures which are more visually straightforward.

Response 3: Grouping of the results after thorough evaluation of all included studies was a very painful procedure. Every table I included consists in detail the results of targeted and untargeted lipidomic analyses separately. My aim was to demonstrate homogeneously all data. If I change a table with a figure, there will be heterogeneity and probably confusion for the readers. What I changed was the style of tables 5, and 6. Also, I adjusted my tables to the format of IJMS.

Thank you in advance.

Round 2

Reviewer 1 Report

The authors have made corrections in the previous manuscript. The current manuscript has been modified with a satisfactory response. 

Reviewer 2 Report

I consider that substantial changes were made with respect to the first version, I have no further comments.